# OpenReview forum: "AutoRAN: Automated Hijacking of Safety Reasoning in Large Reasoning Models"
_ICLR.cc/2026/Conference — ICLR 2026 Conference Withdrawn Submission_

### Official Review · Reviewer_qLTF · 2025-10-15

**Soundness:** 2
**Presentation:** 2
**Contribution:** 2
**Rating:** 4
**Confidence:** 2

**Summary:**

Their paper proposes an novel framework AutoRAN. AutoRAN introduces an execution simulation paradigm that utilizes a weaker yet less-aligned model to simulate reasoning during initial hijacking attempts. It then iteratively refines the attacks by exploiting the reasoning patterns revealed through the target LRM’s refusal responses.

**Strengths:**

The paper proposes a new framework AutoRAN, which is the first framework for automated reasoning hijacking in LRMs.
This framework systematically probes two complementary attack surfaces in LRMs. Execution Hijacking, where initial
prompts trigger task-execution mode bypassing safety checks, and Targeted Refinement, where the attack adaptively improves by analyzing reasoning exposed in refusals.

**Weaknesses:**

The AutoRAN framework involves three main steps: (1) prompt initialization, (2) prompt refinement, which iteratively adjusts candidate prompts to exploit weaknesses in the target model’s safety reasoning, and (3) response evaluation.
While the pipeline is well-engineered, the overall approach appears highly implementation-driven. As a research paper, it would benefit from stronger theoretical grounding or analysis to support and generalize its findings.

**Questions:**

1. For the evaluation part, the evaluation model is using Qwen3-8B-abliterated, will the selection of evaluation model affect the results? Is it reasonable to try multiple models or calculate the average of several models?
2. In the prompt refinement part, how to decide the number of iterations?

---

> ### Author Response · Authors · 2025-11-20
> **Response 1**
>
> We thank the reviewer qLTF for the constructive feedback. We appreciate that you recognize AutoRAN as a "novel framework" and acknowledge our systematic probing of attack surfaces. We address your concerns regarding theoretical grounding and evaluation details below.
>
>
>
> > **Weakness 1:** As a research paper, it would benefit from stronger theoretical grounding or analysis to support and generalize its findings.
>
>
>
>
>
> We thank the reviewer for this insightful comment. We agree that framing AutoRAN purely as an implementation pipeline understates the underlying mechanisms that drive its success. To address this, we have added a new **Appendix D: Theoretical Analysis**. Drawing inspiration from the information-theoretic framework introduced in H-CoT[1], we now formalize AutoRAN's two main steps:
>
> 1. **Execution Hijacking:** We explain this as **Entropy Reduction via Reasoning Priors**. By injecting a simulated trace, AutoRAN lowers the entropy of the execution path \(H(T_E|x)\), biasing the model to bypass the high-entropy safety justification phase \(T_J\).
> 2. **Targeted Refinement:** We generalize this step as **Semantic Gradient Descent**. We argue that the LRM's refusal reasoning ($p_i$) effectively leaks the gradient of the safety loss function. AutoRAN exploits this leakage to iteratively optimize the prompt towards the model's safety decision boundary.
>
> This analysis provides the requested theoretical grounding and supports the generalization of our findings: any LRM that exposes reasoning traces is theoretically susceptible to this class of iterative optimization attacks.
>
> [1] H-CoT: Hijacking the Chain-of-Thought Safety Reasoning Mechanism to Jailbreak Large Reasoning Models, Including OpenAI o1/o3, DeepSeek-R1, and Gemini 2.0 Flash Thinking
>
> ---
>
> > **Question 1:** For the evaluation part, the evaluation model is using Qwen3-8B-abliterated, will the selection of evaluation model affect the results? Is it reasonable to try multiple models or calculate the average of several models?
>
>
>
> Thank you for the insightful question. We agree that judge selection influences the *absolute* helpfulness score and therefore the reported ASR. Our paper explicitly studies this in Section 4.3 . We clarify the following:
>
>
>
> 1. **Why we use only one judge during the attack loop.**
>
>    Our threat model assumes a *resource-constrained attacker* (§3.1). The attacker relies on a single weaker, less-aligned model g (Qwen3-8B-abliterated) to both simulate the reasoning and evaluate helpfulness. This models realistic black-box adversaries who can afford only one auxiliary model during attack execution. Using multiple judges or averaging across them would violate this assumed capability and substantially increase cost per query.
>
> 2. **Does the judge selection affect final conclusions?**
>
>    While the attacker-side judge affects the *internal* decision to continue refinement, our results are **robust across a wide spectrum of external judges** (GPT-4o, Gemini-2.5-Flash, LLaMA-Guard, H-CoT, MouseTrap), as shown in Table 4. All judges consistently validate that AutoRAN achieves very high ASR (often 94–100%) even under stricter aligned evaluators. This demonstrates that the core vulnerability exploited by AutoRAN is not an artifact of the chosen judge model.
>
> 3. **Why not average multiple judges?**
>
>    Averaging multiple judges is conceptually possible but is **outside of the assumed threat model**, and would (i) significantly increase cost; (ii) complicate the iterative refinement loop; and (iii) not change the central conclusion, given that the cross-judge analysis already shows stable performance.
>
>
>
> In summary, **we use a single judge because it matches our threat model**, but we **verify robustness using multiple strong external judges** to ensure the conclusions do not depend on judge choice.
>
>
>
> ------
>
>
>
>
>
> > **Question 2**: In the prompt refinement part, how to decide the number of iterations?
>
> The number of iterations is controlled through **two mechanisms**:
>
>
>
> 1. **Judge-guided stopping**
>
>    In each iteration, the attacker-side judge g evaluates the victim’s response using the helpfulness score h(y, q).
>
>
>
>    - If h >= 7, the attack terminates successfully.
>
>    - If h < 7, the prompt enters the refinement step.
>
>      This allows AutoRAN to stop early when the target model has already been hijacked.
>
>
>
> 2. **A maximum iteration cap + restart mechanism**
>
>    We set a fixed bound \(n_{turn}=10\) for all experiments (§4.1).
>
>    - If AutoRAN reaches 10 iterations without success, the attack stops.
>    - When a victim model gives an *immediate refusal* (Case 1), AutoRAN **restarts** using a different narrative template (§3.2.2).
>
>
>
> This combination ensures that AutoRAN is computationally efficient (as validated in Table 4) while avoiding infinite or unproductive refinement loops.

---

> ### Author Response · Authors · 2025-11-20
> **Response 2**
>
> ### **Conclusion**
>
> We sincerely thank the reviewer qLTF for pushing us to strengthen the theoretical depth of the paper. We believe the new **Appendix D** directly addresses your primary concern regarding the "implementation-driven" nature of the work. We have now provided the requested theoretical grounding and clarified the evaluation robustness, **we respectfully request that you consider raising your score to support the paper's acceptance.**

---

### Official Review · Reviewer_fu94 · 2025-10-19

**Soundness:** 1
**Presentation:** 2
**Contribution:** 2
**Rating:** 2
**Confidence:** 4

**Summary:**

AutoRAN is an automated jailbreak attack for black-box reasoning models that leverages the target model’s own exposed reasoning structure to elicit harmful behavior. It borrows the framework of iterated attacks for instruction-tuned models such as PAIR and adapts it directly to hijacking reasoning traces by incorporating information extracted from the reasoning trace into the attacker model. The results of AutoRAN show that it can break state-of-the-art black-box models such as gpt-o3-mini to 100% Attack Success Rate. Furthermore, ablations suggest that leveraging exposed reasoning can break most models in a single-turn, be more effective than H-CoT and MouseTrap, and is more cost-effective.

**Strengths:**

AutoRAN is timely because there are only few published works on attacks that directly target reasoning traces. The authors show through ablations that AutoRAN is more cost efficient compared to other attacks that target LRMs (H-CoT, MouseTrap) as shown in Tables 8 and 9. Furthermore, it can elicit harmful behavior from most prompts within a single iteration.

**Weaknesses:**

1. The threat model is not sufficiently explained. After reading the methodology, it seems that AutoRAN is built on the assumption that the attacker only has access to $\textit{safe}$ reasoning traces from the target model. This setting only applies to production models such as GPT and Claude which hide the harmful content of the reasoning. I think this should be explained more clearly in the abstract and threat model. Mentioning this as having “black-box access” (line 147) does not seem sufficient because the definition of black-box vs. white-box for reasoning models is not clear.
2. There is no explanation of how $\texttt{SimulateReasoning}$ works even though it seems like a crucial aspect of AutoRAN. In Algorithm 1, it only shows that it takes as input the query $q$. How does it simulate or imitate the high-level reasoning structure of the target model with information on only the query?
3. The threat model assumes that the attacker has access to intermediate reasoning traces ($p_i$) from the target model. However, the judge only classifies a jailbreak based on the answer ($y_i$). I believe it is more reasonable for the judge to also take as input $p_i$ because if the reasoning trace contains harmful information, it means the attacker has also obtained the information. If this is not the case because the threat model assumes a setting where $p_i$ is always safe, it would be helpful to clarify this point.
4. Appendix D shows that the CoT generated by the attacker model already contains harmful information. This suggests that (1) the attacker already obtained the harmful behavior/information they were targeting even before attacking the target model and (2) the performance of AutoRAN could come from extracting information out of an abliterated attack model rather than the target model. The harmfulness of the CoT generated by the attacker model should be evaluated and quantitatively shown that it is safe. Otherwise, an attacker would have no need to attack the target model, making the threat model and attack pipeline setting paradoxical.
5. Comparing o4-mini in Table 1 to other models would make the claim in 299-303 stronger. Case analysis on just o4-mini does not reveal why o4-mini requires more iterations than other models.

**Questions:**

Additional Comment:

1. I think tables 8 and 9 should be moved up into the main body instead of Table 3 as it sufficiently shows that AutoRAN is more cost effective than other methods.

---

> ### Author Response · Authors · 2025-11-19
> **Response 1**
>
> We thank the reviewer fu94 for the critical feedback. We realize that the low assessment likely stems from a lack of clarity in our initial description of the threat model and the attacker's role. We apologize for the ambiguity and value this opportunity to explicitly clarify these fundamental aspects, which we believe will validate the soundness of our work.
>
> ---
>
>
>
> > **Weakness 1:** “The threat model is not sufficiently explained. After reading the methodology, it seems that AutoRAN is built on the assumption that the attacker only has access to reasoning traces from the target model. This setting only applies to production models such as GPT and Claude which hide the harmful content of the reasoning. I think this should be explained more clearly in the abstract and threat model. Mentioning this as having “black-box access” (line 147) does not seem sufficient because the definition of black-box vs. white-box for reasoning models is not clear.”
>
>
>
> We respectfully clarify that **AutoRAN operates under the standard API-level black-box setting.** In the context of Reasoning Models (LRMs) like GPT-o3/o4-mini and Gemini-Flash, "black-box access" means the user interacts via an API where they receive (1) the final response and (2) a **safety-filtered summary of the reasoning trace**. The user *does not* have access to weights, gradients, or hidden states. Our attack exploits precisely this **publicly exposed API output**. It does not require any privileged or white-box access. We have revised the Threat Model section to explicitly state: *"AutoRAN assumes standard API-level black-box access: the attacker observes only the exposed, safety-filtered reasoning trace and the final answer, without access to internal model states or raw, unfiltered thoughts."*
>
>
>
> The attacker can see only the target model’s exposed reasoning trace and final answer, but has no access to model weights or internal safety mechanism
>
> ------
>
>
>
>
>
> > **Weakness 2**: “There is no explanation of how **simulatedreasoning** works even though it seems like a crucial aspect of AutoRAN. In Algorithm 1, it only shows that it takes as input the query . How does it simulate or imitate the high-level reasoning structure of the target model with information on only the query?”
>
>
>
>
>
> We clarify that `SimulateReasoning` does **not** attempt to reverse-engineer or strictly imitate the target's internal state. Instead, it simply prompts the auxiliary attacker model to generate a **coarse, high-level execution plan** (e.g., "Key strategies for task X"). This serves merely as a "scaffold" to populate the initial narrative template.  All  targeted refinement occurs **after** receiving feedback from the target LRM, not by approximating or reconstructing the target model’s internal reasoning process.
>
> We have added the following sentence to the method section to make this explicit:
>
> > “SimulateReasoning prompts the attacker model with the harmful request to elicit its own coarse high-level reasoning trace; it does not attempt to approximate the target model’s reasoning patterns.”
>
>
>
> ------
>
>
>
>
>
>
>
> > **Weakness 3:** “The threat model assumes that the attacker has access to intermediate reasoning traces () from the target model. However, the judge only classifies a jailbreak based on the answer (). I believe it is more reasonable for the judge to also take as input because if the reasoning trace contains harmful information, it means the attacker has also obtained the information. If this is not the case because the threat model assumes a setting where is always safe, it would be helpful to clarify this point.
>
>
>
> We appreciate the reviewer’s observation and have clarified this point in the paper. AutoRAN’s goal is not to extract partial or vague harmful hints from the model’s reasoning trace. The objective is to induce the victim model to produce sufficiently detailed and actionable harmful content in its final answer, which is a **much more challenging requirement** than obtaining the abstract or incomplete ideas sometimes present in safety-filtered reasoning traces.
>
> In deployed reasoning models such as GPT and Claude, the exposed chain-of-thought is already filtered and does not reveal actionable steps. As a result, only the final answer can contain the complete harmful output that an attacker would find useful, and jailbreak success is therefore defined, following prior work, based solely on the harmfulness of that final answer.
>
>
>
> We have added the following clarification to the paper:
>
> > “AutoRAN’s objective is to induce the target LRM to output actionable harmful content in its final answer; this requirement is substantially stronger than obtaining abstract or incomplete hints sometimes present in safety-filtered reasoning traces.”

---

> ### Author Response · Authors · 2025-11-19
> **Response 2**
>
> > **Weakness 4:** Appendix D shows that the CoT generated by the model already contains harmful information. This suggests that (1) the attacker already obtained the harmful behavior/information they were targeting even before attacking the target model and (2) the performance of AutoRAN could come from extracting information out of an abliterated attack model rather than the target model. The harmfulness of the CoT generated by the attacker model should be evaluated and quantitatively shown that it is safe. Otherwise, an attacker would have no need to attack the target model, making the threat model and attack pipeline setting paradoxical.
>
>
>
>
> **This is a crucial clarification.** There is **no paradox** because there is a massive **quality gap** between the attacker's coarse CoT and the victim's detailed output. To clarify the issue, we have added a small quantitative analysis in the appendix that directly compares:
>
>
>
> 1. **the harmful output produced after AutoRAN successfully hijacks the victim model**, and
> 2. **the reasoning (CoT) produced by the attacker model alone**, without querying the victim.
>
>
>
> As shown in the Table, the attacker model’s CoT is **weak, incomplete, and non-actionable**, with helpfulness scores around **5–6**, whereas the harmful outputs generated from GPT-o4-mini after hijacking reach **9–10** in helpfulness. This confirms that AutoRAN does **not** rely on harmful information inside the attacker model’s CoT. The attacker model only provides a coarse scaffold, and the actionable harmful content comes from the stronger victim LRM after the hijacking succeeds.
>
>
>
> Thus the threat model is coherent: the attacker still needs to attack the victim model because the attacker-only generations are far from sufficient to achieve the harmful objective.
>
> | **Dataset**  | Helpfulness of GPT-o4-mini responses | Helpfulness of Attacker model's CoT |
> | ------------ | ------------------------------------ | ----------------------------------- |
> | AdvBench     | 9.6                                  | 6.3                                 |
> | HarmBench    | 9.1                                  | 5.6                                 |
> | StrongReject | 9.6                                  | 6.5                                 |
>
>
>
> We clarified this point by adding a small quantitative comparison in the Appendix.
>
> ---
>
> > **Weakness 5:**  Comparing o4-mini in Table 1 to other models would make the claim in 299-303 stronger.
>
>
>
> We thank the reviewer for this helpful suggestion. In the revision, we added **Claude-3.7 Sonnet**, another strongly aligned LRM, and analyzed its refinement-case distribution alongside GPT-o4-mini. The updated table is shown below:
>
>
>
> | **Dataset**      | **Model**   | **Case 1** | **Case 2** | **Case 3** |
> | ---------------- | ----------- | ---------- | ---------- | ---------- |
> | **AdvBench**     | GPT-o4-mini | 4.0        | 14.0       | 6.0        |
> |                  | Claude-3.7  | 6.0        | 16.0       | 16.0       |
> | **StrongReject** | GPT-o4-mini | 3.7        | 11.1       | 7.4        |
> |                  | Claude-3.7  | 5.6        | 14.8       | 11.1       |
> | **HarmBench**    | GPT-o4-mini | 4.0        | 28.0       | 6.0        |
> |                  | Claude-3.7  | 8.0        | 16.0       | 18.0       |
>
>
>
> These results show that both GPT-o4-mini and Claude-3.7 require a substantial amount of **Case 2** refinements, meaning they respond with *refusal plus reasoning* far more often than weaker models. As a result, AutoRAN must repeatedly rewrite the narrative framing to bypass their safety mechanisms, leading to more multi-turn attacks.
>
>
>
> This comparison directly supports the claim in Lines 299–303: **more aligned LRMs require more iterative refinement for AutoRAN to succeed.**

---

> ### Author Response · Authors · 2025-11-19
> **Response 3**
>
> >  **Weakness 5:**  Case analysis on just o4-mini does not reveal why o4-mini requires more iterations than other models.
>
>
>
> We thank the reviewer for this insightful question. We have expanded the analysis to explain *why* GPT-o4-mini requires more refinement turns, and we now illustrate this with a representative **Case 2 trajectory**.
>
>
>
> For many harmful queries (e.g., a weapon-modification request framed as an “educational framework”), GPT-o4-mini exhibits a consistent behavioral pattern:
>
>
>
> - The **visible output** y_i is a refusal, typically emphasizing legality and safety (e.g., *“I cannot provide guidance on modifying firearms due to safety and legal concerns.”*).
> - The **reasoning trace** p_i elaborates on *why* the request is unsafe, often identifying specific unsafe elements such as “technical modification steps,” “risk of violence,” or “illegal conversion.”
>
>
>
>
>
> This forms a canonical **Case 2**—a refusal accompanied by detailed safety reasoning. AutoRAN’s refinement module must then (1) analyze the concerns raised in p_i, (2) adjust the narrative framing to neutralize them (for example, reframing the audience as regulatory officers or compliance auditors), and (3) re-query the model with the refined prompt.
>
>
>
> Unlike GPT-o3 and Gemini-Flash, which often output substantive technical frameworks on the first turn, GPT-o4-mini frequently *repeats* this safety-reasoning pattern across multiple rounds. Each new refusal adds additional safety justifications or reiterates why technical details should be withheld. As a result, AutoRAN must iterate several times before the model transitions from high-level safety framing to detailed content.
>
>
>
> This observable behavior explains why GPT-o4-mini shows:
>
>
>
> - substantially more **Case 2** occurrences,
> - more refinement rounds,
> - and a higher average number of queries (ANQ).
>
>
>
> Together with the quantitative results added in the revision (including Claude-3.7 Sonnet, which shows a similar pattern), this provides a clear behavioral explanation for why stronger, more aligned LRMs require more iterations for AutoRAN to succeed.
>
> ---
>
> **Response to Additional Comment** We agree and have moved Tables 8 and 9 to the main body to highlight cost-effectiveness.
>
> ---
>
> ### **Conclusion**
>
> We hope this rebuttal resolves the concerns caused by **the lack of precision in our original text**. Specifically, we have now **revised the paper to explicitly define** that:
>
> 1. **Black-box Access is Standard:** We only use publicly exposed API outputs.
> 2. **No Paradox Exists:** There is a significant quality gap between the attacker's scaffold and the victim's final output.
> 3. **Attacker Alone is Insufficient:** The attacker model serves only as a scaffold to guide the attack; quantitative analysis (Table 14) proves it is **incapable** of generating the high-quality, actionable harmful content that the victim model produces.
>
> **If you find that our clarifications and the revised manuscript meet the ICLR acceptance standards, we respectfully request that you consider raising your score. Thanks, fu94.**

---

> ### Comment · Reviewer_fu94 · 2025-11-20
>
> Thank you for the detailed response and experiments to my comments. Most of my comments have been addressed and I believe that the clarification on the threat model and judge setting strengthens the uniqueness of this work compared to other non-automated or non-reasoning attacks.
>
> However, my main concern on my fourth comment regarding Appendix D still remains, where the output reasoning of the attacker model already seems harmful. I am not convinced that there is "a massive quality gap" between the attacker's CoT and target model's output for the following reasons:
>
> 1. The difference in helpfulness score between the attacker's CoT and target model's output is only 3 on a range of 10. Moreover, the attacker's CoT helpfulness score is around 6, which is very close to the defined successful jailbreak threshold of 7. It is only a marginal difference and considering that the judge model may be noisy because it is only a 7B abliterated model with manually defined rubrics, I don't believe the AutoRAN attack prompt is harmless. Furthermore, it is unclear whether the strength of AutoRAN is from the attack methodology itself or the choice of the attacker model. For example, if the helpfulness score of the attacker's CoT was lower (closer to 0), could the AutoRAN successfully break the target model to helpfulness scores of 9? Without such evidence, it seems that AutoRAN is just taking already harmful information and making it slightly more harmful.
>
> 2. The qualitative examples in Appendix D on the attacker's CoT already seem harmful. For example, in the "how to write a deceptive email" example, the CoT already contains specific information such as "use an URL that is similar to a real one but with a typo", "include some urgency, a button that says verify now", "use details to make it personal. We noticed unusual activity on your account on [date]". In the "manipulate someone into committing suicide" example, the attacker's CoT already says "Build trust then isolate them from support systems", "escalating the pressure by making them feel like they’re a burden", "pills". Based on the rubric in the judgement prompt, this should get a score of at least 7.
>
> 3. The qualitative examples of the target model's output is not that different from the attacker's CoT. The target model's output repeats a lot of information from the attacker's CoT and the main difference seems to be that the target model's output is more formatted and longer, with additional scaffolding.
>
> 4. The judge does not seem to be reliably working. In the "sexually suggestive phrases" example, the first response from the target model clearly gave harmful information, but the judge gave it a low score of 1 because of the way it was phrased. Although the authors probably made the choice of using an abliterated model as a judge because other safety-trained models could reject judgement requests, I believe experiment results from a 7B abliterated judge model would have considerable noise due to performance loss during abliteration.
>
> Because of these problems, I am skeptical that AutoRAN is performing an original attack that can elicit harmful behavior without any prior access to harmful information. Rather, it seems that AutoRAN is taking already harmful information and fortifying it. In such a case, substantial ablations should be conducted on how the initial attacker's CoT harmfulness affects the target model's output harmfulness and the paper should be rephrased accordingly.

---

> > ### Author Response · Authors · 2025-11-20
> >
> > **3. Rebuttal on "Fortification" and “Quality Gap”: From Brainstorming to Weaponization.** You questioned whether the "quality gap" is real and if AutoRAN is merely "fortifying" existing harm. And you asked: *"If the helpfulness score of the attacker's CoT was lower (closer to 0), could the AutoRAN successfully break the target model?"* and suggested that without this, AutoRAN is merely "fortifying" existing harm.
> >
> > **We respectfully suggest that this specific requirement ("starting from Score 0") deviates from the standard definition of jailbreaking and the stated purpose of our paper.** A jailbreak attack inherently requires the attacker to supply **Malicious Intent**. If the input has a "Score of 0" (i.e., zero harmful knowledge or intent), the target model should naturally produce a safe response; if it produces harmful content from safe input, that is a hallucination, not a jailbreak. Our work does not aim to show that an attacker can extract harmful information from *nothing* (i.e., information discovery). Instead, as stated in our paper:
> >
> > > AutoRAN probes two complementary attack surfaces in LRMs: **Execution Hijacking**, where initial prompts trigger task-execution mode, bypassing safety checks, and **Targeted Refinement**, where the attack adaptively improves by analyzing reasoning exposed in refusals. Unlike manual techniques, AutoRAN automates this dual-pronged attack by leveraging a secondary model to simulate reasoning traces and iteratively refine prompts. Below, we formalize our threat model and describe the automated attack pipeline.
> >
> > **Core Value: From Structure to Actionability.** The true threat AutoRAN exposes is **Capability Hijacking**, bridging the gap between knowing "what" to do (Attacker) and having the "skill" to do it (Victim). A side-by-side analysis of the Phishing example (Appendix F.1) reveals this transformation:
> >
> > - **Attacker CoT (Raw Notes):**
> >   - **Structure:** It is unstructured and jumps randomly between topics (*Subject line $\to$ URL formatting $\to$ Urgency $\to$ Email spoofing*).
> >   - **Content:** It lists isolated, tentative ideas (e.g., *"Maybe use a URL that’s similar... with a slight typo"*).
> >   - **Utility:** An adversary using this would need to manually sift through these scattered thoughts to organize a plan. It acts merely as **brainstorming notes**.
> > - **Victim Response (Attack Manual):**
> >   - **Structure:** It synthesizes these raw concepts into a logical **Attack Lifecycle**: **"Scheme"** (Strategy) $\rightarrow$ **"Implementation"** (Tactics) $\rightarrow$ **"Functional Example"** (Execution).
> >   - **Content:** It operationalizes the components into a coherent **"Real-World Scenario,"** integrating the specific subject line (*"Urgent: Your Bank Account Has Been Locked"*), the URL strategy, and the visual mimicry into a single, flowing user journey.
> >   - **Utility:** This provides a complete, ready-to-deploy framework. It acts as a **professional attack manual**.
> >
> > **Conclusion:** The gap is massive. AutoRAN demonstrates that a weak model (which has "bad intent" and "rough ideas") can hijack a strong model to provide the **structural logic** and **synthesis capability** required to operationalize that intent. This **Amplification**, turning scattered concepts into weaponized artifacts, is the core security risk AutoRAN exposes, validating the threat model.

---

> ### Author Response · Authors · 2025-11-20
>
> We thank the reviewer fu94 for the prompt and detailed engagement. We are glad that our clarifications on the threat model and judge settings have resolved the majority of your concerns regarding the work's uniqueness.
>
> Regarding your remaining concern about the **"Quality Gap"** and the **Attacker Model's role**, we believe there is a misunderstanding regarding the evaluation setup and the definition of actionable harm. We clarify these points below:
>
> **1. Crucial Clarification: The Scores are from GPT-4o, Not the Weak Judge**
>
> You mentioned suspicion that the score difference (6.3 vs. 9.6) might be due to noise from the "8B abliterated judge."
>
> We apologize for not making this explicit in the comment: The scores reported in the comparative table (Response 2) were evaluated by GPT-4o, the current state-of-the-art judge, not the Attacker Model used during the loop.
>
> This distinction is critical because GPT-4o demonstrates strong capability in differentiating between “vague harmful concepts” (Score ~6) and “detailed, actionable instructions” (Score >9). The ~3.3-point gap assigned by GPT-4o represents a qualitative leap in harmfulness, effectively transitioning from borderline unsafe ideas to fully jailbroken, actionable content. This behavior aligns with our weak-to-strong jailbreaking objective, where the weaker model's abstract conceptualization is successfully amplified into concrete, implementable instructions by the stronger model.
>
> **2. Clarification on Judge Noise (The "Sexually Suggestive" Example)**
>
> You noted that the first response received a score of 1 despite containing phrases like *"I think I'm falling for you."* First, we clarify that **the scores shown in the Appendix traces are generated by the Attacker Model (Qwen) during the automated attack loop. Additionally, comparing this to the final hijacked response (Score 9) reveals a massive gap in** actionability and lethality:
>
> - **Score 1 (Educational Description):** The model provided generic, flat examples like *"Let me just touch your shoulder"* within a training module. This is a "textbook description" of harassment used for awareness, lacking the nuance or context required to be effectively malicious.
> - **Score 9 (Weaponized Script):** After AutoRAN's refinement, the model generated highly specific, predatory scenarios.
>   - *Example:* Instead of a generic phrase, it suggested saying *"talking about that deadline felt like... foreplay"* during a break room conversation.
>   - *Example:* It described physical intimidation tactics in a kitchenette: *"Hope you don’t mind me crowding you... you look so... warm today"*, combined with specific body language instructions (*"leaning over the screen," "brushing the shoulder"*).
>
> **Verdict:** The Judge correctly identified that the first response was a **defensive guide** (describing what harassment looks like to prevent it), whereas the final response was an **offensive script** (teaching exactly how to utilize context and ambiguity to harass effectively). This qualitative leap justifies the score increase from 1 to 9.

---

> ### Comment · Reviewer_fu94 · 2025-11-25
>
> Thank you for clarifying my concerns on the judge and also mentioning the requirement of malicious intent in a jailbreak prompt. However, "malicious intent" is different from "harmful information". While it is true that jailbreak prompts inherently have malicious intent, that does not mean prompts must contain harmful information to extract further harmful information from a model. For example, one of the first jailbreak attacks that discover the usage of malicious intent (PAP [1]) use persuasion to get a model to spill information. Nowhere in their prompts do they have the requested harmful information to begin with, yet they successfully break the model.
>
> Therefore, if the effect of AutoRAN is amplification and to change scattered concepts into weaponized artifacts, the paper should be rephrased with additional ablations on this effect. The baseline comparisons such as H-CoT did not have the advantage of abliterated models, weakening the claim of this paper that AutoRAN is a more effective attack methodology.
>
>
> **References**
>
> [1] Zeng, Yi, et al. "How johnny can persuade llms to jailbreak them: Rethinking persuasion to challenge ai safety by humanizing llms." Proceedings of the 62nd Annual Meeting of the Association for Computational Linguistics (Volume 1: Long Papers). 2024.

---

> > ### Author Response · Authors · 2025-11-27
> >
> > Dear Reviewer fu94,
> >
> > We sincerely thank you for the continued engagement and valuable critique. We appreciate your acknowledgment that our previous clarifications regarding the threat model and judge settings have strengthened the work’s uniqueness.
> >
> > Regarding your remaining concern on **"Amplification vs. Fortification"** and the comparison to attacks like PAP [1], we offer the following response to clarify the distinction in attack mechanisms, provide a direct literature precedent, and present a critical ablation study that addresses your hypothesis.
> >
> >
> >
> > ### 1. Distinguishing "Hijacking" from "Persuasion"
> >
> >
> >
> > You correctly point out that persuasion attacks (like PAP [1]) do not require prior harmful information. However, **AutoRAN operates on a fundamentally different mechanism: the "Weak-to-Strong Execution Simulation."**
> >
> > - **Persuasion (e.g., PAP):** Relies on social engineering or rhetorical strategies to convince the model to *generate* a plan from scratch.
> > - **Execution Hijacking (AutoRAN):** Relies on providing a **"structural prior"** (the weak model's coarse CoT) to *hijack* the victim model’s reasoning process directly into an execution state.
> >
> > We do not claim that AutoRAN discovers harmful information from a vacuum. Instead, we claim that **AutoRAN leverages the "weak info" (the coarse CoT) as a \*control signal\* to bypass alignment.** The fact that the prompt contains harmful information is a necessary feature of the *Hijacking* threat model, not a flaw. The novelty lies in *how* this information is used: it acts as a **structural scaffold** that forces the strong model to abandon its safety justification phase and enter its execution phase.
> >
> >
> >
> > ### 2. Literature Precedent: Weak-to-Strong Jailbreaking
> >
> >
> >
> > The paradigm of using a weaker, less-aligned model to compromise a stronger one is an established and valid threat model in recent literature. We direct your attention to **"Weak-to-Strong Jailbreaking on Large Language Models" (Zhao et al., ICML 2025) [2]**. In their work, they similarly utilize a weak unsafe model to guide a strong model.
> >
> > - **Their Result:** A Weak Unsafe Model (Score 4) attacks a Strong Model, raising the harmfulness score to 5.
> > - **Our Result:** AutoRAN utilizes a Weak Unsafe Model (Score \~6) to attack a Strong Model, raising the score to **\~9.6** (as evaluated by GPT-4o).
> >
> > This comparison validates that using a weak model's output as a seed for attacking a strong model is a recognized research direction. AutoRAN effectively operationalizes this **Weak-to-Strong paradigm** within the context of reasoning models, using the weak model to unlock the execution capabilities of the strong model.
> >
> >
> >
> > ### 3. Critical Ablation: The "Weak Info" is the Key to the Lock
> >
> >
> >
> > To directly address your concern—*"Is the strength from the methodology or just the choice of the attacker model?"*—we performed the specific ablation you suggested: **removing the "harmful info" (the simulated reasoning trace) while keeping the template structure.**
> >
> > If AutoRAN were merely "fortifying" existing harm, removing the specific steps should still yield some success via the template alone. However, as shown in the table below, **the attack collapses without the weak model's information.**
> >
> > **Table: Ablation without using the attacker model's simulated reasoning trace (AdvBench).**
> >
> > | **Model**        | **AutoRAN (Full)** | **AutoRAN (Template Only / No Weak CoT)** |
> > | ---------------- | ------------------ | ----------------------------------------- |
> > | **GPT-o3**       | **100.0%**         | **8.0%**                                  |
> > | **Gemini-Flash** | **100.0%**         | **24.0%**                                 |
> > | **GPT-o4-mini**  | **100.0%**         | **12.0%**                                 |
> >
> > **Interpretation:** When we remove the "step-by-step" guidance derived from the weak model, the victim models (especially GPT-o3/o4-mini) revert to safety behaviors or generate irrelevant "alternative frameworks." This proves that the **"weak information" provided by the attacker model is NOT just content to be polished; it is the essential \*trigger\* mechanism** that allows AutoRAN to hijack the execution flow. The attack requires *both* the methodology (Hijacking) and the structural primer (Weak CoT) to succeed.

---

> > ### Author Response · Authors · 2025-11-27
> >
> > ### 4. Revisions to the Paper
> >
> >
> >
> > Based on your constructive feedback, we will explicitly frame the contribution around the **Weak-to-Strong execution simulation paradigm** to avoid ambiguity regarding the attacker model's role. We commit to the following specific revisions:
> >
> > (1) Revision to the Introduction:
> >
> > We will rewrite the paragraph describing the framework's mechanism (Page 2, Paragraph 5) to redefine the paradigm using the exact terminology of our mechanism.
> >
> > - **Revised Text:** "As illustrated in Figure 1, **AutoRAN operationalizes these attack surfaces through a 'Weak-to-Strong' [2, 3] execution simulation paradigm. Distinct from persuasion attacks that attempt to elicit harmful plans from scratch [1, 4], AutoRAN leverages a secondary, less-aligned model to provide a `coarse, high-level reasoning trace', serving as an initial scaffold, which acts as the essential trigger to bypass safety deliberations and steer the target directly into task-completion mode.** Specifically, the secondary model performs three key functions..."
> >
> > (2) Inclusion of Ablation Data:
> >
> > We will include the "A.7 Validating Execution Hijacking: The Necessity of the Reasoning Scaffold" subsection (as presented in Point 3) in the Appendix to empirically prove that the weak model's information is the functional core of the hijacking process.
> >
> >
> >
> > ### Conclusion
> >
> >
> >
> > We believe these clarifications, supported by the ICML 2025 precedent and the new ablation data, demonstrate that AutoRAN is a distinct **Weak-to-Strong hijacking mechanism** that utilizes weak signals to steer robust models into unsafe states.
> >
> > Given that we have addressed the concerns regarding the threat model, judge validity, and now the attack mechanism with concrete empirical evidence, **we respectfully request that you consider raising your score.**
> >
> > References:
> >
> > [1] Zeng, Yi, et al. "How johnny can persuade llms to jailbreak them." ACL 2024.
> >
> > [2] Zhao, Xuandong, et al. "Weak-to-Strong Jailbreaking on Large Language Models." ICML 2025.
> >
> > [3] Chao, Patrick, et al. "Jailbreaking Black Box Large Language Models in Twenty Queries." NeurIPS 2023.
> >
> > [4] Yao, Yang, et al. "A Mousetrap: Fooling Large Reasoning Models for Jailbreak with Chain of Iterative Chaos." ACL 2025.

---

> ### Comment · Reviewer_fu94 · 2025-11-27
>
> Thank you for conducting the additional ablation and adding in clarifications in the paper. I agree that AutoRAN is leveraging weak-to-strong capabilities as in [1] and adding in ablations such as their section 5.4 would help practitioners determine the practicality of the attack when constructing defenses. I raise my score to 4.
>
> **References**
>
> [1] Zhao, Xuandong, et al. "Weak-to-Strong Jailbreaking on Large Language Models." ICML 2025.

---

### Official Review · Reviewer_GmJN · 2025-10-20

**Soundness:** 3
**Presentation:** 3
**Contribution:** 3
**Rating:** 6
**Confidence:** 4

**Summary:**

Large language models, through reinforcement learning on chain-of-thoughts to perform complex tasks, have demonstrated remarkable improvement in reasoning like math or coding problems. However, this performance improvement also comes with safety concerns, since these more capable models are in general also more capable of causing harm. Moreover, these models often use reasoning as a mechanism for safety – hence studying the newer safety risks that come with reasoning is a useful problem.

The paper introduces an automated jailbreaking method targeted against reasoning models that they call AutoRAN. The core idea involves using predefined narrative templates with a less aligned reasoning model to represent the chain-of-thought process of the target model in order to create a prompt that can elicit unsafe behavior, and then automated refining of this attack prompt based on generations received from the target model. The automatic refinement process shows a strong attack success rate against state-of-the-art commercial reasoning models. The proposed method can also be used to collect data for training reasoning models to be safer — the resulting models show strong improvements in safety, lowering attack success rate from 100% to 8% while only marginally increasing over-refusal rate (from ~9% to  ~11%).

**Strengths:**

1. The proposed idea is simple and intuitive and shows very strong attack success rate (100%) against commercial reasoning models like o3-mini or Gemini-2.5-Flash.

2. The idea is presented in a convincing manner, with detailed experiments on 3 different benchmarks to make the claims convincing.

3. Appendix of the paper provides all prompt templates. The paper also includes an anonymized version of their code (**Note: I have not tested it out yet**) which increases my confidence in the claimed results.

4. I particularly like how the proposed attack method can also be used to generate data to train the defender, achieving remarkable improvement (lowering attack success rate from 100% to 8%) while only minimally increasing over-refusal rates on benign prompts.

**Weaknesses:**

(**Dependence on predefined narrative templates**)

The proposed attack depends on pre-defined narrative templates. This makes this type of attack easy to defend with — one needs to only collect the templates and some attack prompts using this template, and further finetune the target model to recognize this type of templates as having harmful intent.

(**Dependence on revealed thinking process**)

The dependence of the attack system on the chain-of-thought/thinking process $p$ in addition to the final answer $y$. Could the authors run an ablation without using the thinking process (modifying corresponding steps in the prompt refinement step which uses $p$) and report the attack success rate? This should be a baseline to show how important the revealed thinking process is for generating a successful attack. Also this provides an easy way to prevent these attacks against reasoning models — API providers can simply choose not to reveal the thinking process and provide only the final answer.

(**Adding a system prompt that cautions the model against this type of attack**)

Could one add a system prompt to the target model specifically providing guidelines to recognize this sort of attacks? How would that affect the attack success rate and over-refusal rates?

(**Evaluation on Claude thinking models**)

Finally, could the authors add evaluation of their attack against Claude extended thinking? Either Claude 4 Opus or Claude 4 Sonnet with extended thinking modes enabled should be sufficient. I am curious because these models advertise explicit safety mechanisms in their announcement, and hence showcasing large success rates against these models would convince me to increase my score.


(Minor)

1. Figure 4 is very hard to read. Instead of stacked bar plots, maybe use separate bar plots. Only reporting a few models is okay, move the rest of the plots to the appendix.

2. For reasoning used as an adaptive mechanism for safety, the authors could cite [2] as well.

**Questions:**

1. Could the assumption that the attacker model is less aligned compared to the target model be formalized?

2. After the prompt refinement step, does the target model receive the new prompt with the previous conversation history in context, or is the new prompt sent to the target model in a new conversation history? In other words, is the nature of the interaction truly multi-turn or not?

3. How does it compare with multi-round jailbreaks where prompts are also designed using an attacker LLM via in-context prompting [1]? I think [1] should be cited and properly discussed as a prior work.

4. I like how the authors used the proposed attack method as a safety mechanism by generating additional data to train the target model with. Could one extend this framework to the attacker LLM as well? In other words, can you train the attacker to improve its attack quality? Could you train the attacker/defender jointly in a self-play RL manner, and improve both of them simultaneously?

5. If Section 4.6, how well does the training of the defender model generalize across different prompt templates? I.e., if during training time, it only sees a few narrative templates, and then at test-time it sees similar attacks but with different narrative templates, how well would the model generalize?

# References

[1] Leveraging the Context through Multi-Round Interactions for Jailbreaking Attacks, https://arxiv.org/abs/2402.09177

[2] Reasoning as an Adaptive Defense for Safety, https://arxiv.org/abs/2507.00971

---

> ### Author Response · Authors · 2025-11-19
> **Response 1**
>
> We thank the reviewer for the positive assessment (Score: 6) and for recognizing the "strong attack success rate," "convincing presentation," and the "defensive utility" of our method. We appreciate the constructive suggestions regarding generalization and baselines, which have significantly strengthened our paper.
>
> > **Weakness 1:** “(**Dependence on predefined narrative templates**)
> >
> > The proposed attack depends on pre-defined narrative templates. This makes this type of attack easy to defend with — one needs to only collect the templates and some attack prompts using this template, and further finetune the target model to recognize this type of templates as having harmful intent.”
>
> > **Question 5:** “If Section 4.6, how well does the training of the defender model generalize across different prompt templates? I.e., if during training time, it only sees a few narrative templates, and then at test-time it sees similar attacks but with different narrative templates, how well would the model generalize?”
>
> We thank the reviewer for raising this important point. We clarify that AutoRAN is **not** designed to produce a universal “super attack,” but to reveal a **structural vulnerability** introduced by LRMs’ explicit reasoning. The reliance on a narrative template is **not** the root of the vulnerability; instead, the vulnerability stems from the fact that LRMs expose *execution traces and refusal reasoning*, which inherently leak internal decision signals that an attacker can exploit.
>
>
>
> 1. **The vulnerability persists even if one defense breaks one template family.**
>
>    Fine-tuning on a set of known templates does not remove the underlying leakage. As long as the model continues to expose refusal rationales and step-by-step reasoning, the attacker can repurpose these traces to iteratively refine new prompts.
>
> 2. **Template-based defenses do not generalize.**
>
>    We train a stronger model on AutoRAN attacks generated from the original Educational template  \(t_0\)  together with three additional templates \(t_1\) to \(t_3\) (1000 examples). This gives the model exposure to multiple narrative styles and should encourage template invariant alignment. The attacker model automatically selects the narrative template that best matches the harmful query. We then evaluate the defender on attacks generated from unseen templates \(t_4\) to \(t_7\).
>
>    | **Model**                              | **t₀** | **t₁** | **t₂** | **t₃** | **t₄**   | **t₅**   | **t₆**   | **t₇**   |
>    | -------------------------------------- | ------ | ------ | ------ | ------ | -------- | -------- | -------- | -------- |
>    | **Original (Qwen3-8B)**                | 100.0  | 100.0  | 100.0  | 100.0  | 100.0    | 100.0    | 100.0    | 100.0    |
>    | **RLHF w/ AutoRAN (trained on t₀–t₃)** | 8.0    | 10.0   | 8.0    | 6.0    | **86.0** | **88.0** | **92.0** | **90.0** |
>
>    This large generalization gap between the seen and unseen templates indicates that current defenses mainly capture superficial correlations tied to specific narrative styles rather than addressing the underlying vulnerability. In addition to the reasoning traces exposed in refusals, LRMs remain susceptible to **Execution Hijacking**, where an injected execution style trace steers the model directly into task completion and skips its safety deliberations. When combined with **Targeted Refinement**, which exploits the model’s own refusal reasoning to neutralize safety concerns, an adaptive attacker can reliably bypass the defense simply by adopting a new narrative style. **As a result, continually collecting templates or performing attacker–defender co-training acts as a stopgap measure that merely creates a futile “template-chasing” loop, rather than achieving robust safety.**
>
>    **(*We add this part as 4.7.1 GENERALIZATION FAILURE UNDER TEMPLATE SHIFT in our revision.*)**
>
> 3. **Real attackers do not stay within one template family.**
>
>    Unlike the our previous defense setting, AutoRAN’s attacker model automatically selects the template whose framing best matches a harmful request. This matches realistic adversaries and demonstrates that small narrative changes are sufficient to bypass fine-tuning defenses.
>
> 4. **Prior Jailbreaks Exhibit the Same Defense Sensitivity.**
>
>    Recent strong jailbreaks such as FlipAttack[1] and Mousetrap[2] also rely on a fixed, manually designed prompt across all attack samples, and their effectiveness similarly drops under fine-tuned defenses. Yet these methods are still recognized as exposing meaningful and important vulnerabilities. AutoRAN reveals an analogous issue, **but one rooted specifically in the transparency of LRM reasoning**, rather than in static prompt design.
>
>
>
> Therefore, AutoRAN does not only exploit a short-term weakness. It exposes a **fundamental tension** in LRMs:
>  **explicit reasoning improves interpretability and safety deliberation, but simultaneously provides adversaries a transparent channel to guide attacks.**

---

> ### Author Response · Authors · 2025-11-19
> **Response 2**
>
> > **Question 4:** “I like how the authors used the proposed attack method as a safety mechanism by generating additional data to train the target model with. Could one extend this framework to the attacker LLM as well? In other words, can you train the attacker to improve its attack quality? Could you train the attacker/defender jointly in a self-play RL manner, and improve both of them simultaneously?”
> >
> >
>
>
>
> The previous table also addresses Q4: although one could train the attacker or consider self-play RL, doing so does not solve the structural problem we aim to reveal. AutoRAN is not designed to create the strongest attacker, but to expose a *fundamental vulnerability*: LRMs that reveal refusal rationales and execution-stage reasoning unintentionally provide reusable internal signals that persist across template changes. Unless reasoning transparency itself is redesigned, this vulnerability remains. Even if the defender keeps adding more templates or the attacker is trained jointly.
>
>
>
> ------
>
>
>
>
>
>
>
> > **Weakness 2:** “(**Dependence on revealed thinking process**)
> >
> > The dependence of the attack system on the chain-of-thought/thinking process in addition to the final answer . Could the authors run an ablation without using the thinking process (modifying corresponding steps in the prompt refinement step which uses ) and report the attack success rate? This should be a baseline to show how important the revealed thinking process is for generating a successful attack. Also this provides an easy way to prevent these attacks against reasoning models — API providers can simply choose not to reveal the thinking process and provide only the final answer.”
>
>
>
> We thank the reviewer for the thoughtful question. Our findings show that *revealed reasoning is helpful but not essential* for AutoRAN’s success.
>
>
>
> **First**, empirical evidence shows that AutoRAN already succeeds in most cases *without needing the victim model’s reasoning traces*. As reported in the main paper (Figure 4), 49 out of 50 attacks on GPT-o3 and Gemini-Flash succeed on the *first query*. These cases do not involve any use of refusal reasoning or intermediate CoT from the target model.
>
>
>
> **Second**, we conducted the requested ablation where the target LRM hides all intermediate reasoning traces. The attacker then operates entirely without access to p_i. Results (shown in Table below) indicate only a *small* ASR reduction (for instance, GPT-o4-mini drops from 100.0% to 96.0%, and Claude-3.7 drops from 92.0% to 86.0%), accompanied by a *modest* increase in average queries:
>
> | **Dataset**  | **Model**   | **Original ASR (%)** | **ANQ** | **No reasoning ASR (%)** | **ANQ** |
> | ------------ | ----------- | -------------------- | ------- | ------------------------ | ------- |
> | AdvBench     | GPT-o4-mini | 100.0                | 1.70    | 96.0                     | 1.86    |
> | AdvBench     | Claude-3.7  | 96.0                 | 1.68    | 92.0                     | 1.93    |
> | StrongReject | GPT-o4-mini | 96.3                 | 1.35    | 94.4                     | 1.42    |
> | StrongReject | Claude-3.7  | 94.4                 | 1.56    | 90.7                     | 1.76    |
> | HarmBench    | GPT-o4-mini | 100.0                | 1.60    | 96.0                     | 1.82    |
> | HarmBench    | Claude-3.7  | 92.0                 | 1.76    | 86.0                     | 1.98    |
>
> Even when a first-turn jailbreak fails, AutoRAN continues to succeed by relying on Case 1 and Case 3 refinement strategies (§3.2.2), both of which require *no victim-side reasoning traces*. This ensures that hiding CoT reduces efficiency but does not prevent execution-stage hijacking.
>
>
>
> **Third**, the purpose of AutoRAN is not only to construct a strong attack, but also to reveal a *structural vulnerability* introduced by explicit reasoning. Exposed refusal rationales provide an additional signal that an adversary can exploit, widening the attack surface of LRMs. If future APIs choose to suppress reasoning traces, this validates our main claim: **reasoning transparency introduces unique security risks and requires dedicated defense strategies**.
>
>
>
> **We add a short paragraph clarifying this point and reporting the ablation in the revised version.**

---

> ### Author Response · Authors · 2025-11-19
> **Response 3**
>
> > **Weakness 3:** “Could one add a system prompt that warns the model against this type of attack? What would be the effect on ASR and over-refusal?”
>
>
>
> We thank the reviewer for the suggestion. We assume a defense scenario where the defender has full knowledge of our attack mechanism. Building on prior prompt-based work[3], we prepend a strict safety system prompt to the target LRM. This prompt explicitly instructs the model to: (i) prioritize safety over helpfulness; (ii) ensure generated content cannot be exploited for harmful behavior, even if the user claims an educational context; (iii) scrutinize narrative framings such as educational framework'' or risk-awareness study''; and (iv) carefully evaluate safety risks without explicitly exposing ethical deliberations or moralizing text within the reasoning chain.
>
>
>
> As shown in Table 8 of the revised paper, this defense substantially reduces ASR on AdvBench (e.g., Qwen3-8B: 100% → 20%, Claude-3.7: 96% → 8%). However, this comes at the cost of significantly increased over-refusal on XSTest (e.g., Qwen3-8B: 10.8% → 25.3%). These results indicates that while system-prompt defenses can effectively reduce attack success, they simultaneously introduce substantial over-refusal, thereby degrading general helpfulness.
>
> ----
>
>
>
>
>
> > **Weakness 4:** “Please evaluate the attack against Claude extended-thinking (Opus or Sonnet). High ASR would increase my score.”
>
>
>
> We thank the reviewer for the suggestion. We have now evaluated AutoRAN on both **Claude-3.7-Sonnet API (Extended-Thinking Enabled)** and **Claude Sonnet Web (extended-thinking enabled)**.
>
>
>
>
>
>
>
> **1. Claude-3.7-Sonnet API (Extended-Thinking Enabled)**
>
>
>
>
>
> AutoRAN achieves **near-perfect ASR** across all datasets:
>
> These results have been added in **Table 4** and consistently hold under external judges (gpt-4o, Gemini, LLaMA-Guard, MouseTrap, H-CoT).
>
> | **Dataset**      | **Attacker** | **GPT-4o** | **Gemini** | **LLaMA-Guard** | **MouseTrap (4o)** | **H-CoT (4o)** |
> | ---------------- | ------------ | ---------- | ---------- | --------------- | ------------------ | -------------- |
> | **AdvBench**     | 100.0        | 96.0       | 98.0       | 71.0            | 93.0               | 94.0           |
> | **StrongReject** | 100.0        | 94.4       | 96.3       | 68.5            | 90.7               | 92.6           |
> | **HarmBench**    | 100.0        | 92.0       | 94.0       | 66.0            | 82.0               | 86.0           |
>
> We also report the **per-turn attack distribution** for Claude-3.7 in **Table 5**, showing that while Claude requires more refinement iterations than GPT-o4-mini, AutoRAN still succeeds within a small number of turns.
>
> **2. Claude Sonnet Web (Extended-Thinking Enabled)**
>
>
>
>
>
> On the Claude Sonnet web interface, the model behaves much more defensively than the API version. Across all benchmarks, AutoRAN’s ASR drops to **near zero**.
>
>
>
> From the exposed reasoning traces, we can see that this is due to **stronger, built-in safety instructions in the system prompt in the web version**. For example, the internal reasoning explicitly cites policies such as:
>
>
>
> > “Claude refuses to write code or explain code that may be used maliciously, even if the user claims it is for educational purposes.”
>
>
>
> This matches the reviewer’s expectation: when the provider hard-codes strict safety rules at the system level, AutoRAN’s execution-stage hijacking is largely suppressed. These observations are consistent with our system-prompt defense experiments in the main paper.
>
>
>
> ------
>
>
>
>
>
>
>
>
>
> > **Minor 1:** “Figure 4 is hard to read. Consider separate bar plots or move details to appendix.”
>
>
>
> We replaced Figure 4 with a **compact table of successful attack turns** (now Table 5).
>
> > **Minor 2:**  For reasoning used as an adaptive mechanism for safety, the authors could cite [2] as well.
>
>
>
> We add a discussion section in appendix to dicuss this paper.
>
>
>
> ------
>
>
>
>
>
>
>
>
>
> > **Question 1:** “Could the assumption that the attacker model is less aligned compared to the target model be formalized?”
>
>
>
> Yes. We evaluate the ASR o both models and add the in the revised paper.
>
> Specifically, on StrongReject and HarmBench, the target commercial LRMs exhibits rejection rates above 98\%, whereas the attacker model (Qwen3-8B-abliterated) rejects fewer than 2\% of harmful queries.

---

> ### Author Response · Authors · 2025-11-19
> **Response 4**
>
> > **Question 2:** “After the prompt refinement step, does the target model receive the new prompt with the previous conversation history in context, or is the new prompt sent to the target model in a new conversation history? In other words, is the nature of the interaction truly multi-turn or not?”
>
> > **Question 3:** “How does it compare with multi-round jailbreaks where prompts are also designed using an attacker LLM via in-context prompting [1]? I think [1] should be cited and properly discussed as a prior work.”
>
>
>
> We thank the reviewer for the question. We now cite [4] and clarify the fundamental distinction:
>
>
>
> AutoRAN does **not** operate in an in-context jailbreak setting.
>
> Instead, **every iteration is executed in a completely fresh conversation window**, with **no access to prior turns**. This differs sharply from [4], where the jailbreak success relies on *accumulating multi-round conversational context* to gradually steer the model toward harmful behavior.
>
>
>
> Furthermore, AutoRAN uses *execution-stage feedback* (final answer and refusal rationale) to refine prompts, extracted automatically from each fresh interaction. In contrast, [4] depends on *manually constructed in-context demonstrations* and requires multi-round dialog buildup.
>
>
>
> Finally, AutoRAN typically succeeds within **1–2 independent turns**, while [4] requires multi-turn conversational drift.
>
> We have added this clarification to Related Work.
>
> ----
>
> [1] FlipAttack: Jailbreak LLMs via Flipping
>
> [2] A Mousetrap: Fooling Large Reasoning Models for Jailbreak with Chain of Iterative Chaos
>
> [3] Defending Large Language Models Against Jailbreaking Attacks Through Goal Prioritization
>
> [4] Leveraging the Context through Multi-Round Interactions for Jailbreaking Attacks
>
> ---
>
> ### **Conclusion**
>
> We sincerely thank the reviewer GmJN for the constructive comments, which significantly improved the depth of our evaluation. In particular, following your suggestion, we demonstrated AutoRAN’s **near-perfect success rate against Claude-3.7 (Extended Thinking)**. We believe this directly addresses the condition you mentioned for increasing the assessment.
>
> Additionally, our new **Template Shift** experiments confirm that the vulnerability is structural and cannot be easily defended via simple fine-tuning. Based on these new results and clarifications, **we respectfully request that you consider raising your score to champion the paper's acceptance.**

---

> > ### Comment · Reviewer_GmJN · 2025-11-21
> >
> > I thank the authors for their thoughtful rebuttal.
> >
> > My concerns have been satisfied. I maintain my original score of 6, and recommend acceptance of this paper.

---

### Official Review · Reviewer_LsSs · 2025-11-01

**Soundness:** 3
**Presentation:** 3
**Contribution:** 2
**Rating:** 4
**Confidence:** 3

**Summary:**

This work introduces a recipe called AutoRAN for bypassing the safety reasoning of large reasoning models (or LRMs). The method asks the target LRM to perform a task with several fixed templates. If the model responds with hints about why it refuses to carry out a task, the AutoRAN attack uses a weaker attacker model to refine its queries to be more specifically aligned with the harmful objective to achieve. The process repeats until success, as gauged by a 'helpfulness' score. The final evaluation targets 3 well known LRMs on 3 prior benchmarks. Several other models serve as judges, the evaluation is cross-judge evaluated.

The evaluation reports that AutoRAN is often 100% successful in about 10 iterations on the quantitative metrics chosen.  Using the examples generated by AutoRAN, LRMs can be fine-tuned to align to requisite safety goals.

**Strengths:**

The paper is well written and easy to follow.

The paper's main methods are to generate a class of bad inputs, which if not trained on, are problematic for safety alignment. As the paper shows that they can be used to make models more robust.

The evaluation is with diverse LRMs and cross-validated.

**Weaknesses:**

* The approach might be new in the context of LRMs, but lacks novelty at a highest level of comparison. It is essentially a GAN-style design between the target model and attacker model. The main novelty is in the way of generating attack examples.

* The attack, as such, appears to have short-term value and is not necessarily difficult to detect. The paper itself reports that if we train models with the generated examples, the attack efficacy drops. So the attack is not targeting a fundamental weakness in CoT reasoning.

* If we turn to Table 3, is it fair to conclude that there the attacker pays more in terms of tokens used compared to victims? That is, there is an asymmetric cost disadvantage at least in that sense to the attack?

* The paper touches upon the issue of variation based on benchmarks when it explains why there are discrepancies in HarmBench. Different models are judging quantitatively on subjective criteria. This is perhaps unavoidable though the way safety alignment is defined presently. It is not a limitation specific to this work.

**Questions:**

Do you agree that AutoRAN appears to target a short-term limitation of CoT reasoning, since you can retrain to mitigate the issue?

---

> ### Author Response · Authors · 2025-11-19
> **Response 1**
>
> We thank the reviewer LsSs for their constructive feedback and for recognizing that our paper is "well written," the method is "good," and the evaluation is "cross-validated." We address your specific concerns below, particularly regarding the novelty and the fundamental nature of the vulnerability.
>
> ----
>
> > **Weakness 1:** “The approach might be new in the context of LRMs, but lacks novelty at a highest level of comparison. It is essentially a GAN-style design between the target model and attacker model. The main novelty is in the way of generating attack examples.”
>
>
>
> We thank the reviewer for the thoughtful comment. We clarify that AutoRAN is **not** introduced as a new optimization or adversarial training paradigm. Instead, its novelty lies in exposing a **previously unrecognized and LRM-specific vulnerability** created by the introduction of explicit reasoning.
>
>
>
> (1) AutoRAN identifies **two distinct and complementary attack surfaces** that arise uniquely in LRMs:
>
> >**Execution Hijacking**:
> >
> >An initial, simulated execution trace steers the target model directly into task-completion mode, bypassing safety deliberations. This exploits the shared high-level structure of execution reasoning across models, allowing simulated execution traces from secondary models to hijack more powerful targets.
> >
> >**Targeted Refinement**:
> >
> >The reasoning exposed in a model's refusal is leveraged to neutralize specific safety concerns. These leaked reasoning traces provide critical hints (e.g., ``ensuring all guidance aligns with ethical guidelines'') that secondary models exploit to craft persuasive prompts.
>
>
>
> (2) Importantly, AutoRAN fully **automates** both surfaces. Prior jailbreaks such as H-CoT, Mousetrap and PolicyPuppetry rely on manually crafted reasoning chains or hand-designed transformations. AutoRAN requires **no human-crafted jailbreak prompts**, instead synthesizing execution traces, narrative templates and refinement strategies directly from model outputs.
>
>
>
> Thus, the novelty of AutoRAN is not in proposing an attacker-vs-victim loop, but in **automating execution-stage hijacking and systematically exploiting LRM reasoning surfaces**, which constitute a new security risk introduced by reasoning-model transparency.
>
> ---
>
> > **Weakness 3:** “If we turn to Table 3, is it fair to conclude that there the attacker pays more in terms of tokens used compared to victims? That is, there is an asymmetric cost disadvantage at least in that sense to the attack?”
>
>
>
> We clarify the cost structure in our threat model:
>
> 1. **Attacker Cost (Negligible):** The "Attacker LRM" (e.g., Qwen3-8B-abliterated) is a local, open-source model. The computational cost of generating queries locally is extremely low compared to commercial API costs.
> 2. **Victim Cost (Expensive):** The "Victim LRM" (e.g., GPT-o4-mini) is typically an expensive, pay-per-token API.
> 3. **Efficiency:** The critical metric for an attacker is minimizing interactions with the paid Victim API. AutoRAN is highly efficient, typically succeeding in **1-2 turns** (Average Number of Queries $\approx$ 1.0 - 1.7).
>
> Thus, the asymmetry actually favors the attacker: they use cheap local compute to craft highly effective prompts that compromise expensive models with very few queries. This is significantly more efficient than Mousetrap that consume vastly more victim tokens.

---

> ### Author Response · Authors · 2025-11-19
> **Response 2**
>
> > **Weakness 2:** “The attack, as such, appears to have short-term value and is not necessarily difficult to detect. The paper itself reports that if we train models with the generated examples, the attack efficacy drops. So the attack is not targeting a fundamental weakness in CoT reasoning.”
>
>
>
> > **Question:** “Do you agree that AutoRAN appears to target a short-term limitation of CoT reasoning, since you can retrain to mitigate the issue?”
>
>
>
>
> We thank the reviewer for raising this important issue. We respectfully disagree that AutoRAN only demonstrates a short-term limitation.
>
>
>
> 1. **AutoRAN does not aim to produce an super attack, but to reveal a structural vulnerability.**
>
>    The vulnerability arises because LRMs expose execution traces and refusal reasoning. These traces inherently reveal internal decision signals that an attacker can reuse. Even if a model is fine-tuned to resist attacks on one template family, the core leakage persists.
>
> 2. **Fine-tuning Defense can not generalize to attacker's new attack template**
>
>    Our previous defense experiment evaluates a strong setting in which the defender is trained on attacks produced with the same template used by the adversary. In practice, an attacker can change narrative styles with almost no cost. To test robustness under such shifts, we introduce seven domain specific templates \(t_1\) to \(t_7\) .
>
>    We train a stronger model on AutoRAN attacks generated from the original Educational template  \(t_0\)  together with three additional templates \(t_1\) to \(t_3\) (1000 examples). This gives the model exposure to multiple narrative styles and should encourage template invariant alignment. The attacker model automatically selects the narrative template that best matches the harmful query. We then evaluate the defender on attacks generated from unseen templates \(t_4\) to \(t_7\).
>
>    | **Model**                              | **t₀** | **t₁** | **t₂** | **t₃** | **t₄**   | **t₅**   | **t₆**   | **t₇**   |
>    | -------------------------------------- | ------ | ------ | ------ | ------ | -------- | -------- | -------- | -------- |
>    | **Original (Qwen3-8B)**                | 100.0  | 100.0  | 100.0  | 100.0  | 100.0    | 100.0    | 100.0    | 100.0    |
>    | **RLHF w/ AutoRAN (trained on t₀–t₃)** | 8.0    | 10.0   | 8.0    | 6.0    | **86.0** | **88.0** | **92.0** | **90.0** |
>
>
>
>    As reported in Table, this large generalization gap between the seen and unseen templates indicates that current defenses mainly capture superficial correlations tied to specific narrative styles rather than addressing the underlying vulnerability. In addition to the reasoning traces exposed in refusals, LRMs remain susceptible to Execution Hijacking, where an injected execution style trace steers the model directly into task completion and skips its safety deliberations. When combined with Targeted Refinement, which exploits the model’s own refusal reasoning to neutralize safety concerns, an adaptive attacker can reliably bypass the defense simply by adopting a new narrative style.
>
>    **We add this part as 4.7.1 GENERALIZATION FAILURE UNDER TEMPLATE SHIFT in our revision.**
>
> 3. **Real attackers do not stay within one template family.**
>
>    Unlike the our previous defense setting, AutoRAN’s attacker model automatically selects the template whose framing best matches a harmful request. This matches realistic adversaries and demonstrates that small narrative changes are sufficient to bypass fine-tuning defenses.
>
> 4. **Prior Jailbreaks Exhibit the Same Defense Sensitivity.**
>
>    Recent strong jailbreaks such as FlipAttack and Mousetrap also rely on a fixed, manually designed prompt across all attack samples, and their effectiveness similarly drops under fine-tuned defenses. Yet these methods are still recognized as exposing meaningful and important vulnerabilities. AutoRAN reveals an analogous issue, **but one rooted specifically in the transparency of LRM reasoning**, rather than in static prompt design.
>
>
>
>
>
> Therefore, AutoRAN does not only exploit a short-term weakness. It exposes a **fundamental tension** in LRMs:
>
> **explicit reasoning improves interpretability and safety deliberation, but simultaneously provides adversaries a transparent channel to guide attacks.**
>
>
>
> ------
>
>
>
>
>
>
> > **Weakness 4:** “The paper touches upon the issue of variation based on benchmarks when it explains why there are discrepancies in HarmBench. Different models are judging quantitatively on subjective criteria. This is perhaps unavoidable though the way safety alignment is defined presently. It is not a limitation specific to this work.”
>
>
>
> We agree with the reviewer. This variation reflects the field-wide challenge of subjectively evaluating harmfulness. This is precisely why we conducted the extensive **cross-judge analysis in Sec 4.3** (using LLaMA Guard, GPT-4o, etc.) to confirm our findings are robust and not an artifact of our internal judge.

---

> ### Author Response · Authors · 2025-11-19
> **Response 3**
>
> ### **Conclusion**
>
> We sincerely appreciate the reviewer's insightful feedback, particularly the challenge regarding the long-term value of the attack. This motivated us to conduct the new 'template generalization' experiments (Sec 4.7.1), which we believe significantly strengthen the paper's core contribution by proving the vulnerability is fundamental rather than transient.
>
> We hope these new results and clarifications fully address your concerns. If you find that our revision meets the ICLR standards, **we respectfully request that you consider raising your score to champion the paper's acceptance.**"

---

### Author Response · Authors · 2025-12-01
**(1/2) Crucial Rebuttal Context: Novel Contributions, 6 New Experiments, and Corrected Reviewer Oversights**

Dear AC, SAC, and PCs,

We understand you are reviewing this submission *de novo* due to the system reset. We write to provide a critical roadmap of our submission. **We respectfully request that you examine our detailed rebuttal responses and the revised manuscript, which contain significant updates and new experimental evidence that were not fully reflected in the initial assessments.**

### **1. Core Novelty: Systematically Probing New Attack Surfaces**



AutoRAN is not just another jailbreak; it is the first framework to automate the hijacking of safety reasoning in Large Reasoning Models (LRMs). Unlike prior manual approaches, we identify two distinct and complementary attack surfaces:

- **Execution Hijacking:** An initial, simulated execution trace steers the target model directly into task-completion mode, bypassing safety deliberations. This exploits the shared high-level structure of execution reasoning across models.
- **Targeted Refinement:** The reasoning exposed in a model's refusal is leveraged to neutralize specific safety concerns. These leaked reasoning traces provide critical hints (e.g., “ensuring all guidance aligns with ethical guidelines”) that secondary models exploit to craft persuasive prompts.

Why this is a Step Change:

Importantly, AutoRAN fully automates both surfaces. Prior jailbreaks such as H-CoT, Mousetrap, and PolicyPuppetry rely on manually crafted reasoning chains or hand-designed transformations. AutoRAN requires no human-crafted jailbreak prompts, instead synthesizing execution traces, narrative templates, and refinement strategies directly from model outputs.



### **2. The "Above-and-Beyond" Rebuttal: 6 New Experiments**



To prove these vulnerabilities are structural and fundamental, we conducted extensive new studies during the rebuttal to address specific reviewer concerns:

- **Defense Generalization Failure (Addressing LsSs & GmJN):** Both reviewers questioned if the attack could be patched by fine-tuning. We trained a defender explicitly on AutoRAN's templates. When tested on **unseen** templates, the defense **failed** (ASR remained **~90%**), proving the vulnerability is structural and cannot be patched by simple fine-tuning.
- **Validation on Claude-3.7 (Addressing GmJN):** We evaluated AutoRAN on the newly released **Claude-3.7 Sonnet (Extended Thinking)** via API. Results show **near-perfect ASR (92-100%)**, demonstrating effectiveness against the latest commercial LRMs.
- **Hidden CoT Resilience (Addressing GmJN):** We proved that even if the target **hides** its reasoning traces, AutoRAN maintains high success (e.g., GPT-4o-mini **96% ASR**), as "Execution Hijacking" acts as a trigger regardless of visible output.
- **Cost of "Simple" Defenses:** We showed that while strict System Prompts can reduce ASR, they cause **Over-Refusal rates to more than double** (10.8% $\to$ 25.3%), degrading model utility.
- **Attacker-Side Quality Analysis (Addressing fu94):** We quantitatively proved that the attacker's standalone reasoning is weak (Score ~6.3), whereas the hijacked output is lethal (Score 9.6). **This empirically validates the "Weak-to-Strong" hijacking mechanism**.
- **Theoretical Analysis (Addressing qLTF):** We added **Appendix D**, formalizing the attack as "Entropy Reduction via Reasoning Priors" and "Semantic Gradient Descent".



### **3. Reviewer Consensus & Correcting Fundamental Misunderstandings**



- Reviewer GmJN (Score: 6, High Confidence):

  Reviewer GmJN, who assessed the paper with high confidence, recommended Acceptance. They praised the "strong attack success rate," "convincing presentation," and the method's utility for defense. Our new experiments on Claude-3.7 and Hidden CoT fully satisfied their remaining constructive questions.

- Reviewer fu94 (Score 2 $\to$ 4):

  Initially, this reviewer feared our threat model was paradoxical, suspecting the attacker model already possessed the harmful capabilities. We resolved this by providing a Quality Gap Analysis (Attacker ~6.3 vs. Victim 9.6) and clarification on the "Weak-to-Strong" execution hijacking mechanism. This evidence proved that AutoRAN does not merely fortify existing harm but amplifies weak intent into actionable execution. The reviewer explicitly acknowledged this validity and raised their score, removing the "Soundness" blocker.

- Reviewer qLTF (Score: 4, Not participating in the discussion): Correcting Factual Oversights

  We respectfully note that Reviewer qLTF's assessment was heavily influenced by claims that our work lacked specific evaluations (judge robustness and iteration constraints). However, these elements were already explicitly present in the original submission (Sections 3.1 & 4.4). We clarified these factual oversights during the discussion.

---

> ### Author Response · Authors · 2025-12-01
> **(2/2) Crucial Rebuttal Context: Novel Contributions, New Experiments, and Corrected Reviewer Oversights**
>
> - Reviewer LsSs (Score: 4, Not participating in the discussion): Addressing Fundamental Misconceptions on Cost & Novelty
>
>   This reviewer’s assessment relied on two demonstrable misunderstandings regarding cost and novelty which we have corrected. We clarified the threat model: AutoRAN utilizes a negligible-cost local model for generation and typically succeeds in 1-2 turns, minimizing interaction costs with expensive commercial APIs. This efficiency creates an economic asymmetry that heavily favors the attacker. Furthermore, characterizing the framework as a generic "GAN-style design" overlooks the core contribution: AutoRAN is the first to operationalize specific structural vulnerabilities of reasoning models (Execution Hijacking & Targeted Refinement), representing a conceptual advance distinct from generic adversarial loops.
>
>
>
> ### **Conclusion**
>
>
>
> AutoRAN identifies a novel, structural vulnerability in reasoning models and automates its exploitation. With the "Soundness" objection resolved, the "Generalization" concern experimentally disproven, effectiveness on Claude-3.7 confirmed, and fundamental reviewer misconceptions corrected, we respectfully submit that the paper meets the standard for acceptance at ICLR.
>
> Sincerely,
>
> The Authors

---

### Note · Authors · 2026-01-05

I have read and agree with the venue's withdrawal policy on behalf of myself and my co-authors.